# Photocatalytic Self-Cleaning Cotton Fabrics Coated by Cu_2(_OH)PO_4_ under VIS/NIR Irradiation

**DOI:** 10.3390/ma12020238

**Published:** 2019-01-11

**Authors:** Dawei Gao, Lili Wang, Chunxia Wang, Tan Chen

**Affiliations:** 1Key Laboratory for Advanced Technology in Environmental Protection of Jiangsu Province, Yancheng Institute of Technology, Yancheng 224051, China; monvlily2008@126.com (L.W.); cxwang@mail.dhu.edu.cn (C.W.); 15365765595@163.com (T.C.); 2College of Textiles and Clothes, Yancheng Institute of Technology, Yancheng 224051, China

**Keywords:** cotton fabric, Cu_2_(OH)PO_4_, photocatalysis, self-cleaning property

## Abstract

In the present work, a mild strategy was employed to obtain cotton fabrics (CFs) coated with Cu_2_(OH)PO_4_ (CHP) nanoparticles to achieve self-cleaning property. The phytic acid (IP6) assisted method was employed to synthesize nanoparticles (CHP-IP6). The as-prepared coated cotton fabrics were characterized using the following techniques: Fourier-transform infrared spectroscopy (FTIR), X-ray diffractometry (XRD), scanning electron microscopy (SEM), energy-dispersive X-ray (EDX) analysis, transmission electron microscopy (TEM), X-ray photoelectron spectroscopy (XPS), and thermogravimetric analysis (TGA). The CHP-IP6 coated cotton fabrics showed significant photocatalytic activity, excellent photocatalytic stability, and good discoloration of methylene blue (MB) stains when exposed to sunlight, which could have important applications as tablecloths, household apparels, and industrial workwear.

## 1. Introduction

Shortage of energy and environmental pollution are two major concerns of the 21st century, posing a threat to human society and economic development, particularly in developing countries [1]. Hence, there is an urgent need to find renewable energy sources to replace fossil fuels [2]. Compared to many other resources, solar energy is clean and inexhaustible and causes no pollution. Therefore, it has gained increased attention and can be used either directly or indirectly [3]. Human activities that consume and contaminate large quantities of water are increasing sharply. There is, therefore, an urgent need to effectively reduce water consumption and prevent water pollution.

It is difficult to recycle and separate traditional catalysts from reaction systems. Immobilization of these catalysts onto solid supports can help to overcome this problem [4]. Cotton fabrics, which are made from renewable resources and used as supporting material, are characterized by good moisture absorptivity, air permeability, durability, and biodegradability. Self-cleaning technology can degrade contaminants or remove organic materials from fabrics into wastewater. High catalytic efficiency, low cost, easy recycling, and environmental sustainability make such technology very promising and attractive [5,6].

Extensive efforts have been made world-wide to design highly-efficient photocatalysts. Nanostructured TiO_2_ with different morphological forms such as nanoparticles, nanowires, nanofibers, or nanorods, has superior photocatalytic properties. It has, therefore, gained tremendous attention in the past few years. However, the wide band gap of these catalysts reduces efficiency and limits its use in solar-energy [7,8,9,10]. Hence, there is an urgent need to develop active and stable visible-light-responsive and even near-infrared-driven photocatalysts. Among the photocatalysts, Cu_2_(OH)PO_4_ has gained special importance primarily due to its unique photocatalytic properties [11,12,13]. Recent studies have shown that CHP is a near-infrared-activated photocatalyst, capable of directly generating photo-induced carriers under NIR irradiation. Additionally, CHP has proven to be a promising Fenton reagent, owing to its unique structure. However, CHP photocatalysts reported in previous studies are difficult to recycle, which limit their further development [14,15].

In the present work, an economical and ‘green’ reagent, phytic acid (IP6) served as a soft template for the preparation of CHP. Next, a mild, versatile, and green method was employed to immobilize CHP onto the cotton fabrics. Subsequently, the photocatalytic activities of CHP and CHP-coated cotton fabrics for the degradation of MB under visible light and infrared light irradiation were also studied. Furthermore, the self-cleaning property of CHP coated cotton fabrics under sunlight was also evaluated by monitoring the removal of MB stains.

## 2. Experimental

### 2.1. Materials and Reagents

White-colored plain weave 100% cotton fabric (50 mm × 50 mm, 120 g m^−2^), purchased from Yancheng Kaiyuan Textile Co., Ltd. (Yancheng, China), was washed with hot deionized water several times and dried at 60 °C for 2 h. (3-Aminopropyl) triethoxysilane (KH550) was purchased from Nanjing Chuagshi Chemical Company (Nanjing, China). All the reagents, dopamine (DA), glutaric dialdehyde, copper (II) nitrate trihydrate (Cu(NO_3_)_2_·3H_2_O), disodium hydrogen phosphate (Na_2_HPO_4_·2H_2_O), phytic acid (IP6), and methylene blue (MB), were purchased from Sigma Aldrich (Shanghai, China). All chemicals used in this work were of analytical grade and were used without further purification.

### 2.2. Methods

#### 2.2.1. Preparation of Cu_2_(OH)PO_4_ Photocatalyst

Cu_2_(OH)PO_4_ photocatalyst was synthesized by a facile hydrothermal process [16]. The steps involved in the synthesis are shown schematically in Figure 1. In a typical procedure Cu(NO_3_)_2_·3H_2_O (2 mmol) was dissolved in 45 mL water, and IP6 solution (0.005 mol L^−1^, 5 mL) was added to the above solution. Next, Na_2_HPO_4_·2H_2_O (1 mmol) was added under vigorous stirring and the solution was heated to a boil. Then, the reaction mixture was transferred to a Teflon-lined stainless-steel autoclave (100 mL) and heated at 120 °C for 8 h. Subsequently, the products obtained were rinsed with ethanol and water at least three times each, and dried in a vacuum oven at 60 °C for 12 h. For comparison, Cu_2_(OH)PO_4_ photocatalyst was prepared by the same method, however, without adding IP6. The samples prepared with and without IP6 were labeled as CHP-IP6 and CHP, respectively.

#### 2.2.2. Fabrication of Cu_2_(OH)PO_4_ Photocatalyst-Coated Cotton Fabrics

First, the cotton fabrics (5 cm × 5 cm), having a specific surface area of 2.2 m^2^ g^−1^ and pore volume of 0.0122 cm^3^ g^−1^ (Appendix A), were immersed into a solution containing ethanol (45 mL), deionized water (2.5 mL), and KH550 (2.5 mL) and stirred constantly for 2 h. Next, the cotton fabrics were dried at 60 °C for 2 h. Then, the as-prepared Cu_2_(OH)PO_4_ photocatalyst (0.1 g) was dispersed in aqueous DA solution (0.2 mg mL^−1^, 50 mL) under constant stirring for 1 h. Subsequently, the solution mixture was centrifuged and washed with deionized water three times. Finally, the Cu_2_(OH)PO_4_ photocatalyst and fabrics were added to a 10% glutaraldehyde solution under vigorous stirring. After stirring for 2 h, the fabrics were washed with deionized water and the sample was designated as CHP-IP6/Cotton and dried at 60 °C for 60 min.

### 2.3. Materials Characterization

X-ray diffraction (XRD) patterns of the samples were obtained on X-ray diffractometer (Bruker, D8-advance, Karlsruhe, Germany), which used monochromatic CuKα (λ = 0.15418 nm) source and 2θ range of 10 to 80. Specific surface areas and pore size distributions were calculated by Brunauer-Emmett-Teller (BET) and Barrett–Joyner–Halenda (BJH) methods, respectively (Beckman Coulter, Fullerton, CA, USA). FTIR spectra of uncoated and coated cotton fabrics were recorded by IRPrestige-21 spectrometer (Shimadzu, Kyoto, Japan). The morphologies of the prepared samples were studied by scanning electron microscopy (SEM, JSM-6700F, JOEL, Tokyo, Japan) as well as transmission electron microscopy (TEM, Philips CM120, Amsterdam, the Netherlands). X-ray energy dispersive spectroscopy (EDS, Oxford X-Max, Oxford Instruments, Thames, UK) in conjunction with SEM was employed to verify the components of CHP-IP6/Cotton composites. The surface chemical composition of CHP-IP6/Cotton composites was analyzed by X-ray photoelectron spectroscopy (XPS, Quantum 2000, Physical Electronics, Chanhassen, MN, USA). The concentration of Cu(II) ions after each test cycle was analyzed using atomic absorption spectrometry (TAS-990AFG, Beijing Purkinje General Instrument Co. Ltd., Beijing, China). Diffuse reflectance spectra (DRS) of the samples were recorded using a UV–VIS–NIR spectrophotometer (U4100, Hitachi, Tokyo, Japan), with an integrating sphere attachment, in the wavelength range of 200 to 2500 nm and BaSO_4_ as the standard reference. The self-cleaning properties of the fabrics were quantified by *K*/*S* values tested on a bench top color spectrophotometric system (Color-Eye 7000A, X-Rite, Grand Rapids, MI, USA). Thermogravimetric analysis (TGA, Q500, TA, USA) was conducted using nitrogen as the purging gas, in the temperature range of 30–600 °C at a heating rate of 10 °C min^−1^, to study the decomposition behaviors of the coated cotton fabrics.

### 2.4. Photocatalytic Activity

The photocatalytic activities of the samples were evaluated from the results of photodegradation of methylene blue (MB) at ambient temperature (25 °C) under visible light and infrared radiations. Each sample (50 mg) was added to aqueous MB solution (60 mL), which had an initial concentration of 20 mg L^−1^. The solution (pH = 6.4) was magnetically stirred in dark for 30 min to attain adsorption-desorption equilibrium. Next, 30% hydrogen peroxide solution (1.0 mL) was added. Subsequently, the photodegradation was performed using a 300 W xenon lamp, provided with a 420 nm or 800 nm cut-off filter. The concentration of MB in the solution at given time intervals was determined using UV-Vis spectrophotometer (Hitachi, UV-1080, Tokyo, Japan) at a wavelength of 665 nm. The degradation efficiency was defined in terms of C/Co, where Co denotes the concentration of MO after attaining equilibrium, and C is the concentration at different time intervals after irradiation. For the composites of CHP-IP6/cotton, the fabric was fixed on a stainless-steel bracket, one side of the fabric facing upwards and the other side facing downwards. Then, aqueous MB solution (5 mg L^−1^, 60 mL) with 30% hydrogen peroxide (1.0 mL) was used for the photocatalytic evaluation.

The effects of relevant active species, such as hydroxyl radicals (^•^OH), superoxide radicals (^•^O_2_**^−^**), and holes (h^+^) were investigated by adding 1 mM tert-butanol (t-BuOH), 1 mM p-benzoquinone (BQ) or 1 mM ethylenediamine tetraacetic acid disodium salt (EDTA-2Na) solutions into the MB solution before adding the photocatalyst. The experiment was carried out in a manner similar to the photodegradation experiment.

### 2.5. Self-Cleaning Property

Based on the discoloration of MB (1 g L^−1^), the self-cleaning properties of the samples were assessed. One drop of MB was dropped onto each fabric sample (2.5 cm × 2.5 cm). The stained specimens were irradiated using solar light on a clear day (520 ± 30 W m^−2^, 28 ± 4 °C, 56–86% RH). Additionally, they were all photographed at different time intervals.

## 3. Results and Discussion

### 3.1. Raman and FTIR Analysis

Raman spectra of the as-prepared CHP-IP6 powders are shown in Appendix A. Bands due to OH^−^ lattice vibrations were observed at 816 cm^−1^. The Raman bands at 976 cm^−1^ and 453 cm^−1^ were attributed to the *ν*_1_ and *ν*_2_ modes of vibrations of phosphate, respectively. Bands due to the *ν*_3_ vibration mode was observed at 1020 cm^−1^, whereas those due to *ν_4_* modes were at 646, 627, and 557 cm^−1^. Peaks attributed to –OH stretching modes, symmetric stretching vibrations of PO_4_^3−^, and vibrations of Cu–O bond were observed in the infrared spectra of the CHP-IP6 powders (Appendix A), also confirming the formation of pure copper hydroxyphosphate [17].

The FTIR spectra of pristine and coated cotton fabrics are shown in Figure 2. The broad absorption bands at 3340 cm^−1^ and 3275 cm^−1^ in the spectrum of the coated fabrics were assigned to the O–H stretching vibrations of cotton. The reduction in intensities of these peaks in the spectrum of CHP-IP6 coated fabrics confirmed the reduction of these groups on the surface of fiber. This could also be confirmed from the reduction of peaks at 1158.0 cm^−1^ and 1107.0 cm^−1^, which were attributed to the C–O stretching vibrations of primary and the secondary alcohols, respectively. The absorption peak at 1052.9 cm^−1^, ascribed to the *ν_2_* vibration mode of the phosphate group was slightly increased [18]. Absorption band at 994 cm^−1^ represented the P–O stretching vibrations of PA [19].

### 3.2. X-ray Diffraction (XRD) Analysis

Figure 3 shows that for both, the pristine and coated cotton fabrics, the characteristic peaks corresponding to crystalline cellulose structure appeared at 2θ = 14.8, 16.7, 23.1, and 34.6° [20]. The characteristic peak of CHP in the coated fabrics appearing at 2θ = 30.17° could be indexed to (220) of JCPDS no. 36-0404.

The XRD pattern (Appendix A) shows a highly crystalline structure of as-prepared samples. The characteristic diffraction peaks of CHP and CHP-IP6 were consistent with JCPDS file no. 36-0404, with the chemical formula Cu_2_(OH)PO_4_, crystallized in an orthorhombic system with the lattice constants of *a* = 8.43 Å, *b* = 8.08 Å, and *c* = 5.90 Å.

### 3.3. Scanning Electron Microscopy (SEM) Images and Energy Dispersive Spectroscopy (EDS) Analysis

Typical FESEM and TEM images of the as-prepared products are shown in Appendix A. SEM shows the morphology of CHP (Appendix A) as microrods with 150–300 μm length and 20–50 μm width. The diameter and length of cotton fiber are 10–20 μm and 25–35 cm, respectively. CHP-IP6 (Appendix A) appeared as quasi-uniform flake-like structures with 30–100 nm width (Appendix A). The morphological properties of untreated and CHP-IP6 coated fabrics are shown in Figure 4a,b, respectively.

Compared to CHP-IP6/cotton fabric, the cotton fabric had a smooth surface, before the coating process (Figure 4a). The surface of the fabrics coated with CHP-IP6 was relatively rough and was composed of many particles, as shown in Figure 4b. This proved that CHP-IP6 was successfully coated onto the fabric surface. In Figure 4c, C, O, Cu, and P elements could be found in the coated fabrics and these findings are consistent with the SEM analysis.

### 3.4. X-ray Photoelectron Spectroscopy (XPS) Analysis

The surface composition and chemical states of Cu, P, and O atoms in CHP-IP6/cotton fabrics can be seen in Figure 5a–d. The survey spectrum (Figure 5a) shows photoelectron peaks for copper (Cu 2p), oxygen (O 1s), silicon (Si 2p), and carbon (C 1s). This suggested that the CHP-IP6 particles were successfully grafted onto the surface of cotton fabrics. It was evident from Figure 5b that photoelectrons of Cu 2p_1/2_ and Cu 2p_3/2_ spin orbitals were located at binding energies of 956.5 eV and 936.5 eV, respectively, suggesting of Cu^2+^ states in the sample. Besides, peaks at ca. 963.9 and 944.2 eV were characteristic of samples that contained Cu^2+^ ions with d^9^ configuration in the ground state [17]. The peak at 134.3 eV in Figure 5c corresponded to P 2p, suggesting that all the P atoms had valence states of +5. The O 1s peaks showed binding energies of 531.4 and 533.2 eV in Figure 5d. The peak at 533.2 eV was associated with the Cu–O bond in CHP, while the other peak at 531.4 eV could probably be due to adsorbed oxygen.

### 3.5. UV–VIS–NIR Diffuse Reflectance Spectroscopy

The optical absorption properties of pristine and coated cotton fabrics are shown in Figure 6. The pristine cotton fabrics showed relatively weak absorptions, especially in the ultraviolet region. However, the peaks of the coated cotton fabrics were significantly large in the entire range of wavelength. Moreover, CHP-IP6 coated cotton fabrics showed stronger photoresponse, compared to the CHP coated fabrics. The UV–VIS–NIR absorption spectrum of Cu_2_(OH)PO_4_ samples, derived from its diffuse reflectance measurement is shown in Figure 6. The CHP-IP6 coated sample showed very broad absorption bands with higher intensity. It is noteworthy that there was no obvious absorption above 1500 nm. The NIR absorption of CHP-IP6 could be fit into four Gaussian peaks centered at 670, 864, 1121, and 1349 nm. The absorption beyond 2000 nm was attributed to the lattice and OH stretching modes [21].

### 3.6. Photocatalytic Performance

The photodegradation efficiencies of CHP and CHP-IP6 coated fabrics under VIS–NIR irradiation are shown in Appendix A. The CHP-IP6-coated fabric showed very high efficiency and could degrade MB by 62% in just 5 min under VIS–NIR irradiation. Compared to CHP, the degradation efficiency for CHP-IP showed 1.3 times increase over 20 min. The transient photocurrent responses of CHP- and CHP-IP6-coated fabrics are shown in Appendix A. As the results suggest, CHP-IP6 showed a less significant improvement in the photocurrent response than CHP. The CHP-IP6 or its coated fabrics showed good photocatalytic activities, which was primarily due to increased surface area, stronger absorption strength, as well as the higher photocurrent response of CHP-IP6. Meanwhile, BET measurements showed that CHP-IP6 had a larger specific surface area (3.05 m^2^ g^−1^), which was 1.7 times higher than that of the CHP. Trapping experiments were performed to explore the major active species of the catalyst in the photocatalytic process. Therefore, as for photocatalysts in the present work, the enlarged specific surface area improved the photocatalytic activity. This was due to the large increase in the number of reactive sites and shortened bulk diffusion lengths, which reduced the recombination probability of photoexcited charge carriers. Both CHP-IP6 and CHP displayed a typical type-III isotherm with an H3 hysteresis loop according to BDDT classification (Figure 7a,b), indicating the presence of a number of mesopores. Moreover, the hysteresis loop of CHP-IP6 shifted to higher P/Po values, indicating the existence of macropores [22,23,24].

The degradation efficiency of CHP-IP6 coated fabrics showed higher photocatalytic efficiency (ca. 99%) in 30 min in Figure 7a. The photolysis contributed immensely to the degradation of dye while adsorption had only minor effects on the dye removal, which could be seen during the photocatalytic degradation of MB (Appendix A). The photocatalytic stability of CHP-IP6 coated fabrics was evaluated from the degradation experiments of MB under VIS–NIR irradiation, which was repeated five times. Figure 7b showed 6.4% decrease in photocatalytic activity for degradation of MB. Moreover, the low concentration of Cu^2+^ ions (0.126 ppm), obtained after five consecutive cycles, suggested a high stability of the CHP-IP6 coated fabrics in the photocatalytic process for MB degradation. Additionally, the catalytic properties of coated fabrics were tested under infrared irradiation (Figure 7c). The photodegradation ratios of MB over CHP and CHP-IP6 coated fabrics were 40% and 72%, respectively, after 12 h under NIR irradiation. This showed that the sample had certain catalytic activity under infrared light.

### 3.7. Thermogravimetric Analysis (TGA) Analysis

Taking into account the conditions for use or maintenance of the fabrics and determination of CHP-IP6 content, thermogravimetric analysis was conducted to evaluate the thermal stability and degradation behavior of the fabrics (Figure 8). The TGA curve could be divided into three decomposition regions, namely, primary, core, and char [25]. In the primary region between 30–150 °C, mass loss of the cotton fabric occurred due to evaporation of water from the amorphous regions. The core region, corresponding to the single stage degradation process of polymer chain, showed a sudden drop between 250 and 380 °C for all the samples.

In the final stage, complete decomposition of polymeric chains occurred at 400 and 600 °C for pristine and coated cotton fabrics, respectively. The CHP-IP6 coating had improved the thermal stability of cotton fabrics. The residue for the coated fabrics was 18.7 wt.%, whereas in the case of pristine cotton it was only 8.5 wt.%.

### 3.8. Self-Cleaning Properties

The images of stain degradation on fabrics after 3, 6, 9, and 18 h of exposure to sunlight in the self-cleaning test are shown in Figure 9. It was clear from the figure that the stains on pristine fabrics still remained after irradiation for 12 h. Both CHP- and CHP-IP6-coated cotton fabrics showed good catalytic effect for stain degradation, which proved the excellent photocatalytic property of the coated fabrics. Compared to CHP coated fabrics (1.1 m^2^ g^−1^), the color of stains on CHP-IP6 fabrics (1.7 m^2^ g^−1^) was much lighter for the same time interval, which showed the consistent effect of photocatalysis presented earlier. The color strength of stained fabrics remained consistent with the *K*/*S* values of the samples, when exposed to sunlight for different time periods, which is directly proportional to the amount of dye present on it (Appendix A).

## 4. Conclusions

In conclusion, novel CHP-IP6/CFs were successfully prepared by a mild and facile method. CHP-IP6 nanoparticles were also prepared and then coated effectively onto the surface of the cotton fibers. When exposed to either VIS–NIR or NIR radiation, both CHP and CHP-IP6-coated fabrics showed high catalytic efficiency. Furthermore, the CHP-IP6/CFs-coated fabric showed excellent sunlight-driven self-cleaning properties, suggesting their brighter application prospects in tablecloths, household apparel, and industrial workwear.

## Figures and Tables

**Figure 1 materials-12-00238-f001:**
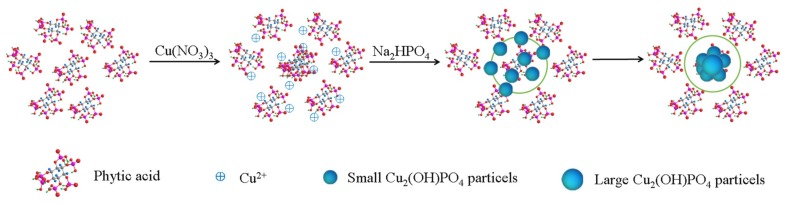
Schematic representation of the process for the synthesis of Cu_2_(OH)PO_4_ photocatalyst.

**Figure 2 materials-12-00238-f002:**
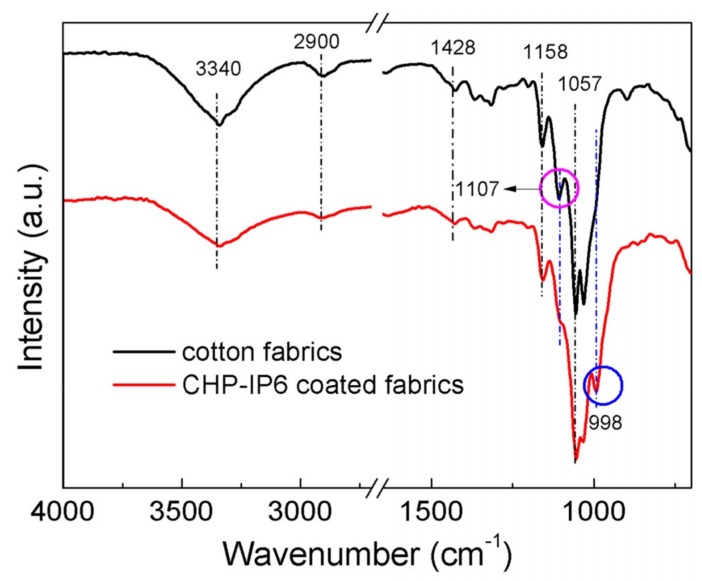
Fourier-transform infrared spectroscopy spectra of pristine and coated cotton fabrics.

**Figure 3 materials-12-00238-f003:**
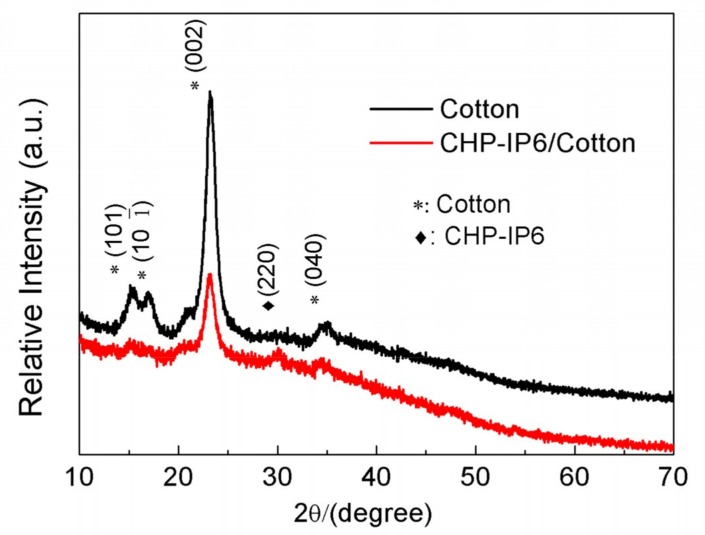
X-ray diffractometry patterns of cotton fabric and CHP-IP6 coated cotton fabric.

**Figure 4 materials-12-00238-f004:**
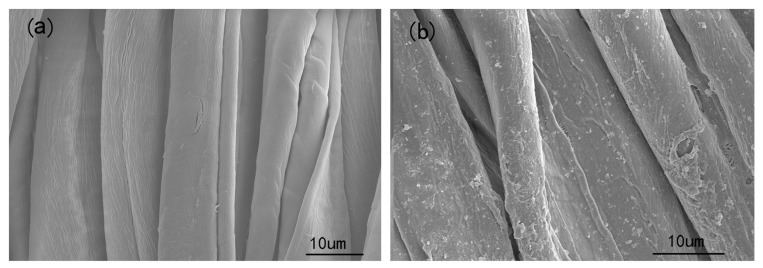
Scanning electron microscopy images of (**a**) cotton fabrics, (**b**) CHP-IP6 coated cotton fabrics, and (**c**) EDS of CHP-IP6-coated cotton fabrics.

**Figure 5 materials-12-00238-f005:**
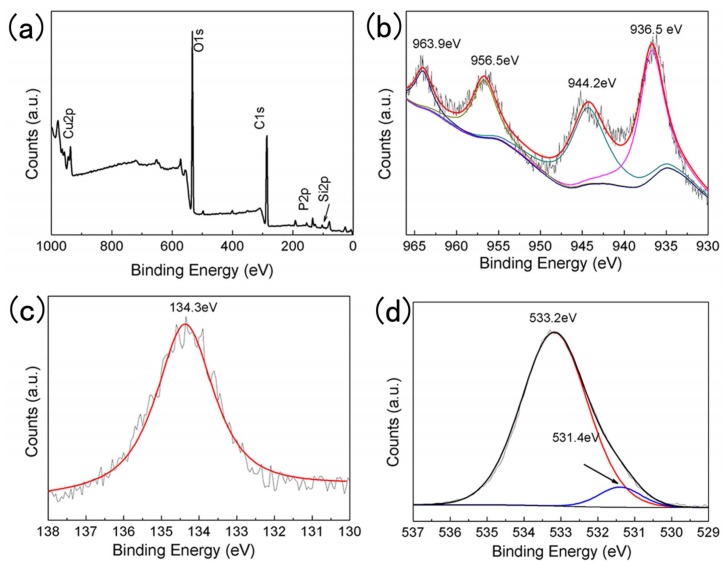
X-ray photoelectron spectroscopy spectra of the coated cotton fabrics, (**a**) survey spectrum; (**b**) Cu 2p; (**c**) P 2p; and (**d**) O 1s.

**Figure 6 materials-12-00238-f006:**
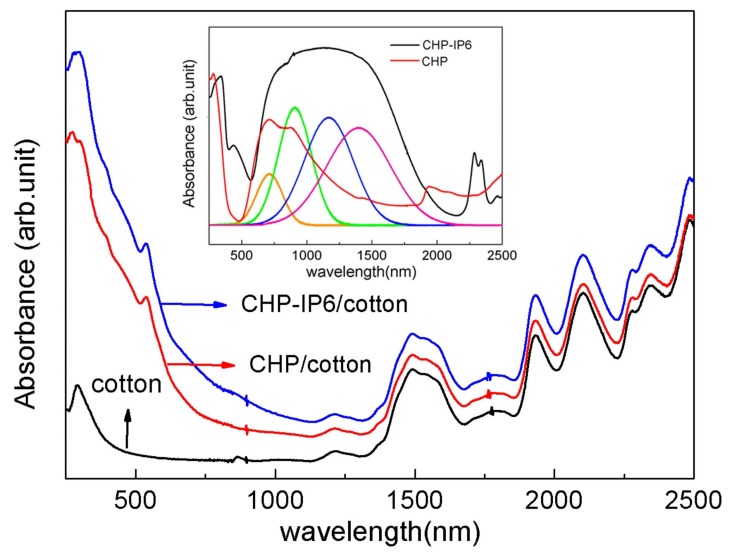
UV–VIS–NIR diffuse reflectance spectra of pristine cotton and coated cotton fabrics.

**Figure 7 materials-12-00238-f007:**
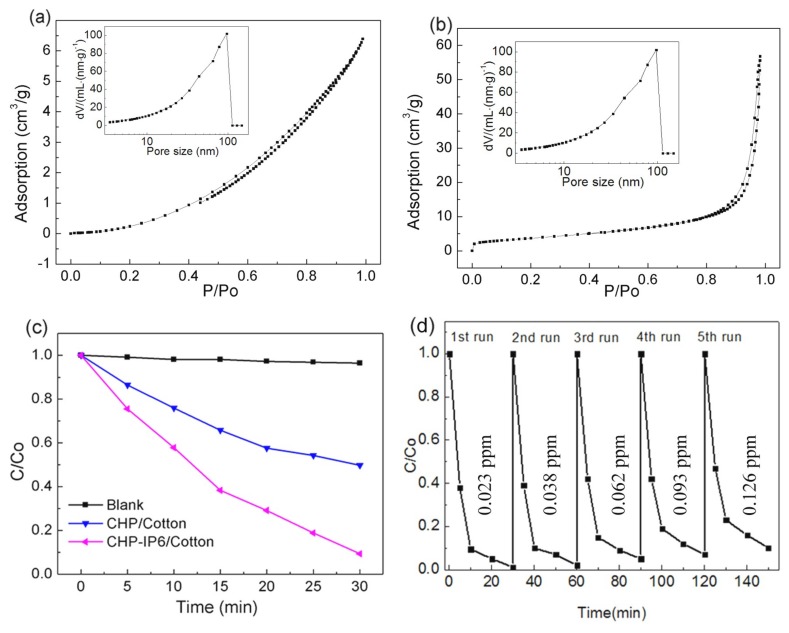
Nitrogen adsorption-desorption isotherms and pore size distributions of (**a**) CHP and (**b**) CHP-IP6; (**c**,**e**) Photocatalytic degradation of MB under VIS–NIR and NIR irradiation, respectively; and (**d**) cycling tests under VIS–NIR irradiation and concentrations of Cu(II) ions after each test cycle.

**Figure 8 materials-12-00238-f008:**
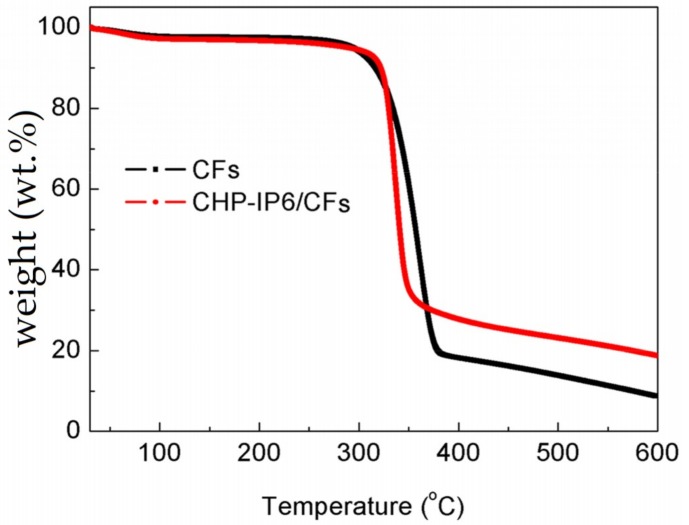
Thermogravimetric analysis curves of pristine cotton and coated fabrics.

**Figure 9 materials-12-00238-f009:**
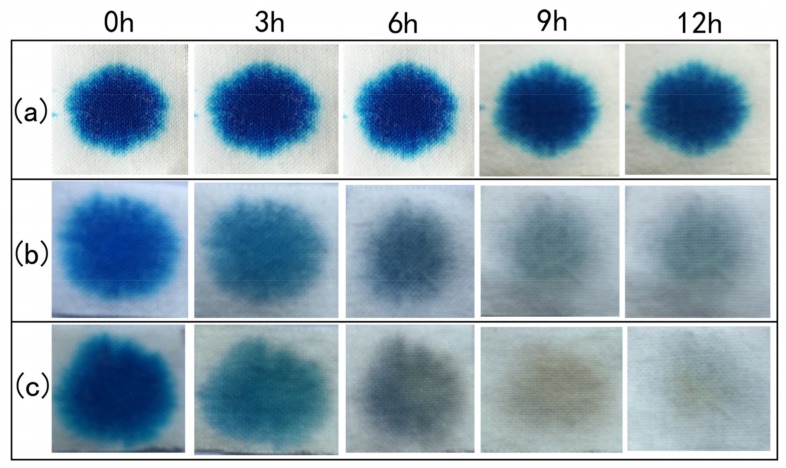
Images of photocatalytic degradation of MB stains on pristine (**a**) CHP, (**b**) CHP-IP6, and (**c**) cotton fabrics on exposure to solar irradiation.

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
