# Peer review of "Photocatalytic Self-Cleaning Cotton Fabrics Coated by Cu2(OH)PO4 under VIS/NIR Irradiation"

_materials, 2019, doi:10.3390/ma12020238_

Round 1
Reviewer 1 Report
Please find the attachment.

Author Response
Dear Sir/Madam,
Enclosed please find our revised manuscript entitled “Photocatalytic self-cleaning cotton fabrics coated by Cu2(OH)PO4 under Vis/NIR irradiation" for consideration for publication in Materials. A detailed point-by-point set of responses to the reviewer’s comments have been provided. Please see the attached word file.
Thanks for your consideration of this revision and hope that the correction will meet with approval.
Yours sincerely,
Dawei Gao

Reviewer 2 Report
I have some doubts about the correctness of photocatalytic analysis. Therefore, I would like to ask Authors for an explanation, what do the authors understand under the term blank test? Whether it is a dark reaction or photolysis process? At the same time, I want to point out that both tests must be carried out to explain the effect of photolysis and adsorption on the effectiveness of dye removal, especially under visible light.
The stability of copper ions in the structure of photocatalyst during the photocatalytic reaction should be also measure by for example leaching of copper ions analysis using atomic absorption spectroscopy. Authors should also report in the experimental section the pH value and the temperature of the solution used for the photocatalytic tests.
Author Response

(The authors gave the same response as above.)

Reviewer 3 Report
The authors aimed to present the obtaining and characterization of photocatalytic self-cleaning cotton fabrics coated by Cu2(OH)PO4 under Vis/NIR irradiation. This manuscript shows serious flaws, English language errors and additional experiments are needed. I cannot recommend its publication in Materials without major revisions.
- The Abstract is very short. It should be stated that cotton fabrics were coated with nanoparticles, not « to make nanoparticles on cotton fabrics». It should end with a phrase regarding the application of these modified fabrics.
- The introduction does not provide the previous results that are already reported in the literature for the CHP nanoparticles attached on fabrics.
- You did not provide the details regarding the cotton fabric that you have used in this study.
- How did you score the self-cleaning efficiency?
- I recommend to introduce TEM images of the obtained nanoparticles.
- Please discuss the results comparing with the literature available.
- What are the possible applications of these results? Please include them in the Conclusion section in order to show the applicability of these results.
- The English language must be strongly revised throughout all manuscript. Also, please follow the guidelines for this journal.
Author Response

(The authors gave the same response as above.)

Round 2
Reviewer 1 Report
Authors answered majority of the comments. The quality of the manuscript improved tremendously after taking care of reviewer's comments. However, it is highly recommended that the authors should include the BET data-graphs of CHP-IP6 and CHP coated CF in the manuscript -
Line: 223- "The CHP-IP6 or its coated fabrics showed good photocatalytic
activity, which was primarily due to increased surface area, stronger
absorption strength as well as the higher photocurrent response of
CHP-IP6."- In context of the above statement, the authors should
explain more clearly how the surface area helped in improvement of the photocatalytic performance. Please include N2- adsorption-desorption isotherm and pore size
distribution graphs of CHP-IP6 and CHP in the main manuscript.
Also, - Line 226:" Meanwhile, BET measurements showed that CHP-IP6 had a larger
specific surface area (3.05 m2 g-1), which was 1.7 times higher than that of the CHP."
- Line 82: "First, the cotton fabrics (5 cm × 5 cm, ), having a specific surface area of 2.2 m2 g-1 and pore volume of 0.0122 cm3 g-1 (Figure S6)"
Did the author measure BET surface area of CHP and CHP-IP6 coated cotton fibers? If yes, please include the data. It is suggested that the authors can compare the surface areas of the above coated cotton fibers as the better photo activity of CHP-IP6 coated fibers was attributed to their increased surface area.
Figure 7(c)- Double check- CHF/Cotton ? Should be CHP/.cotton
Author Response

(The authors gave the same response as above.)

Reviewer 3 Report
The paper can be accepted after minor revision. Please make corrections to methodological errors and text editing.
Author Response
Dear Sir/Madam,
Enclosed please find our revised manuscript entitled “Photocatalytic self-cleaning cotton fabrics coated by Cu2(OH)PO4 under Vis/NIR irradiation" for consideration for publication in Materials. A detailed point-by-point set of responses to the reviewer’s comments have been provided. Please see the attached word file.
Thanks for your consideration of this revision and hope that the correction will meet with approval.
Yours sincerely,
Dawei Gao
Note: All modified places in the revised manuscript are marked in red according to you suggestions. The language was polished more carefully this time and the changes are marked in blue.
Reviewer 2:
Comments and Suggestions for Authors: The paper can be accepted after minor revision. Please make corrections to methodological errors and text editing.
Response: Thanks for your suggestions. We have revised the manuscript more carefully this time and the corresponding revisions of manuscript were highlightened in blue.